# Study on the Horizontal Distribution Law of Flood Water and Sediment Factors under the Effect of Vegetation on a Curved Beach

**Mingwu Zhang [1], Pan Li [1], Xiaoping Li [1], Aoxue Wang [2], Zhenhai Wang [2] and Shengqi Jian [2,*]**

1  Yellow River Institute of Hydraulic Research, Yellow River Conservancy Commission, Zhengzhou 450003, China; thuzmw08@126.com (M.Z.); blondepan@126.com (P.L.); gff1232021@126.com (X.L.)
2  College of Water Conservancy Science and Engineering, Zhengzhou University, Zhengzhou 450001, China; wangaoxuer@163.com (A.W.); wangzhenhai@stu.zzu.edu.cn (Z.W.)
*  Correspondence: jiansq@zzu.edu.cn

**Abstract:** The sediment-laden floodplain flood is affected by beach vegetation and the shape of curved compound channels. The laws of water and sediment exchange and deposition distribution in beach troughs are very complex and play a significant role in the formation and development of secondary suspended rivers, the adjustment of beach horizontal gradients, and even the evolution of flood control situations. This study used a combination of experimental simulations and theoretical research to carry out a generalized model test of floodplain flood evolution, analyzing the transverse distribution characteristics of sediment-laden flow and sediment factors in a curved compound channel under the conditions of beach vegetation, proposing a theoretical model of transverse distribution of velocity and sediment concentration that is based on the momentum equation considering the inertial force of the lateral secondary flow and river curvature. The results showed the following: (1) The model test results for floodplain flood in the compound channel with curved vegetation showed that the main stream was not only concentrated in the main channel but also appeared near the foot of the left and right bank levees and formed flood discharge along the embankment, as the beach siltation was mainly concentrated in the beach lip; (2) The arrangement of full vegetation on the beach had a uniform effect on the velocity distribution of the beach, which can reduce the phenomenon of excessive velocity at the foot of the beach and increase the velocity effect in the main channel; and (3) Through five numerical examples, the lateral velocity distribution model of a curved compound channel with beach vegetation was tested and, in general, the analysis model was consistent with the experimental results. The research results will provide a theoretical basis for river management and have great significance for enriching the basic theory of water and sediment movement and promoting the integration of hydraulics, river dynamics, and ecology.

**Keywords:** beach vegetation; curved compound channel; sediment-laden flow; transverse distribution

## 1. Introduction

A natural river is usually composed of a main channel through the main stream and a beach that carries additional flow. The main channel adjusts itself to form a certain degree of curvature due to factors such as upstream water and sediment and the composition of the riverbank, and the beach land often has different types of vegetation due to the long-term non-flow [1]. Study on the law of water and sand movement in bends is the basis of the study on the law of water and sand movement in natural rivers. Under the action of gravity and centrifugal force, the curved water flow will produce unique motion characteristics that are mainly manifested as the vertical and horizontal gradient of the water's surface, the transverse circulation, the redistribution of the vertical average velocity, etc. It is a complex three-dimensional spiral turbulent flow that makes the characteristics of flow and sediment movement in curved channels different from that in straight channels. The characteristics

of flow in a curved channel determine the basic laws of sediment movement and riverbed evolution, which have an important impact on riverbed evolution, river regulation, bank protection, etc. [2,3].

Beach vegetation has a significant impact on the flow and sediment movement. On the one hand, when the water flows over the beach, vegetation blocks the water flow, reduces the flow velocity in the vegetation area and its vicinity, and changes the water flow structure, which leads to the occurrence of a strong transverse shear layer and affects the exchange of river mass and momentum, thus affecting the overall water transport capacity of the river channel and causing sediment deposition. On the other hand, vegetation plays a certain role in environmental beautification, water ecological restoration, soil and water conservation, and bank consolidation [4–6].

The research on the structure of floodplain water and sediment under the effect of vegetation in the meandering compound channel was mainly focused on the movement law of clear water flow and bed load. As for sandy rivers, sediment-laden flow is one of the characteristics that cannot be described by the existing clear water flow and bedload movement rules in terms of physical movement and sediment transport characteristics. It is necessary to improve the existing theoretical model and establish a lateral distribution model of water and sediment transport under the coupling effect of curved complex channels and beach vegetation, which is suitable for sediment-laden floodplain flow [7,8]. In addition, due to the addition of "beach vegetation" and a "curved compound river channel", it is very difficult to study the influence of sediment-laden floodplain floods on riverbed shape adjustment [9,10].

Although the existing research has some limitations, it also proves that under certain environmental conditions, the influence of the growth of riparian vegetation on the flow resistance is smaller than previously thought [11,12]. Due to the limitation of field measurement, the research on the flow structure of river with vegetation is mainly carried out in indoor flume. The existing indoor research results show that the flow structure in the vegetated area is far more complex than we realize [13]. Due to the limitation of the previous measuring instruments, the early research mainly focused on the estimation of the average velocity, resistance law, and roughness coefficient, and the flow structure in the vegetation area has not been described, analyzed, and understood in detail [14]. The significance of these hydrodynamic characteristics in flood control and sediment movement is not fully studied. For the compound section channel with vegetation on the beach, although the surface shear stress generated by the momentum exchange between the beach and channel flows has been calculated by most researchers on the basis of the unbalanced force, the technology of measuring the surface shear stress directly by using pulsating function is not perfect [15–17]. In addition, the method of quantifying the momentum transfer at the interface between the main channel and the beach by using measurable parameters and considering the surface shear stress of the beach vegetation needs to be developed [18]. Up to now, the experimental study of flow in vegetated channel, whether it is single-section channel or compound-section channel, is only limited to straight channel, without considering the influence of section shape on flow [19–21]. Therefore, it is necessary to study the bend flow with vegetation.

In this study, a curved compound channel model of beach vegetation was established by means of experimental simulation, analysis of measured data, and theoretical research. Based on the established model, the generalized model test of suspended mass floodplain flood evolution was carried out. The transverse distribution characteristics of flow and sediment factors in a curved compound channel under the influence of beach vegetation are analyzed, and an analytical solution model for the transverse distribution of velocity and sediment concentration was proposed. This achievement will provide the theoretical basis for beach area application and river regulation and has great significance for enriching the basic theory of water and sediment movement, promoting the integration of hydraulics, river dynamics, and ecology.

## 2. Materials and Methods

*2.1. Experimental Design*

2.1.1. Model Design

The total length of the curved compound generalized channel model was 60 m, the total width of the cross section was 7 m, and the width of the main channel was 70 cm. The inner radius of the 120° arc in the bending section was 150 cm, and the outer radius was 220 cm. The walls on both sides of the flume were brick cement walls, the bed surface was made of fly ash and the bed surface gradient was 0.2%. In order to control the water level in the tank, an adjustable electric tail door was set at the tail of the tank, and the water from the tail of the tank flowed into the underground reservoir (Figure 1).

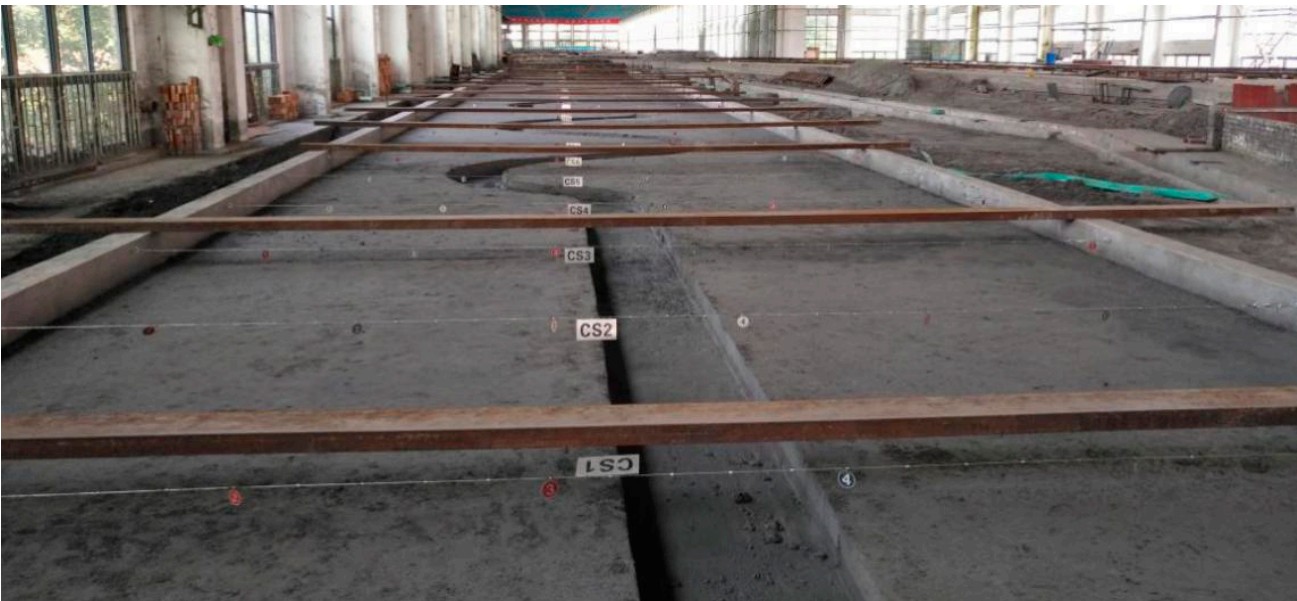

**Figure 1.** Panorama of the generalized water tank model.

The experiment considered the vegetation arrangement in five cases: (1) a curved compound channel with non-vegetation on the beach, (2) curved compound channel with vegetation on the convex bank of the beach, (3) curved compound channel with vegetation on the concave bank of the beach, (4) curved compound channel with vegetation on both sides of the beach, and (5) curved compound channel full of vegetation on the beach. Disposable chopsticks (22.5 cm in length and 5 mm in diameter) were used to simulate vegetation. The planting height of the floodplain vegetation was 7 cm, and the diameter was 5 mm. Concerning the vegetation layout cases (1)–(4): the horizontal distance between the vegetation and the main trough was 10 cm, the vertical distance between the vegetation was 4 cm and the velocity measurement sections were CS6 and 8 (curved compound channel with non-vegetation on the beach), CS10 and 12 (curved compound channel with vegetation on the convex bank of the beach), CS14 and 16 (curved compound channel with vegetation on the concave bank of the beach), and CS18 and 20 (curved compound channel with vegetation on both sides of the beach). For case (5): the horizontal and vertical distances between vegetation were 10 cm, and the velocity measurement sections were CS22 and CS24 (Figure 2).

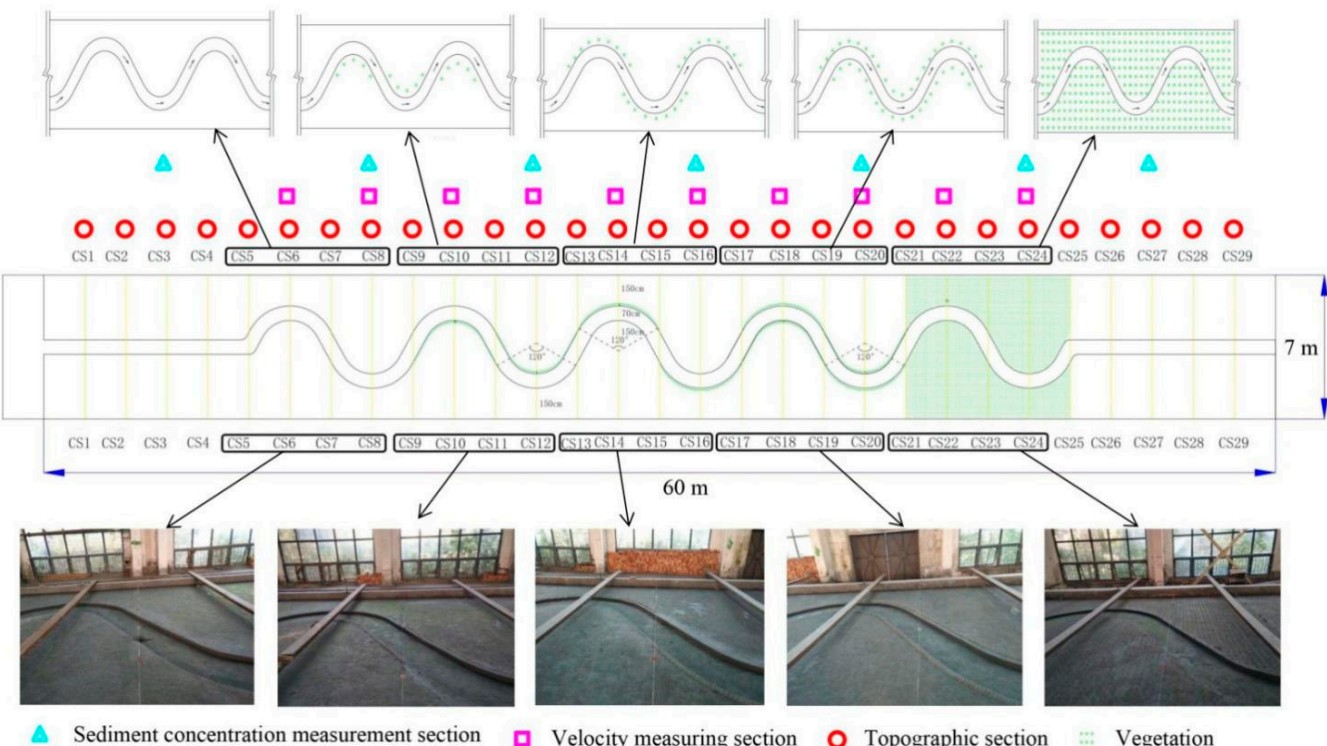

**Figure 2.** Experimental design. (CS5–CS8, curved compound channel with non-vegetation on the beach; CS9–CS12, curved compound channel with vegetation on the convex bank of the beach; CS13–CS16, curved compound channel with vegetation on the concave bank of the beach; CS17–CS20, curved compound channel with vegetation on both sides of the beach; and CS21–CS24, curved compound channel full of vegetation on the beach).

### 2.1.2. Experimental Arrangement

In this study, the LS300-A portable flow meter was used to measure the flow rate; the pycnometer method was used to measure the sand content. The measuring instruments included pycnometers and high-precision balances; sediment particle size was measured by a sediment laser particle size analyzer.

In order to study the flow and sediment situation of curved compound channels with vegetation on the beach more comprehensively, seven types of working conditions were designed through different combinations of sediment concentration and sediment particle size. The design flow was 100 m³/h, and the actual situation was a little different. Condition 0 was the initial condition to adapt to the flow conditions of the initial design terrain; condition 1 was under the condition of clear water; conditions 2–4 were under the condition of relatively fine sediment; and conditions 5–7 were under the condition of relatively coarse sediment. The different sediment concentrations were approximately 5 kg/m³ for small sediment concentrations, 14.5 kg/m³ for medium sediment concentrations, and 35.3 kg/m³ for large sediment concentrations (Table 1). The test tailgate can automatically adjust the height and control of the water level of the tailgate.

**Table 1.** Test condition.

| Working Condition | Design Flow (m³/h) | Actual Flow | Design Sediment Concentration | Actual Sediment Concentration(kg/m³) | Design Particle Size |
|---|---|---|---|---|---|
| 0 | 100 | 100.2 | no | 0 | no |
| 1 | 100 | 90.7 | no | 0 | no |
| 2 | 100 | 93.1 | small | 5.23 | fine |
| 3 | 100 | 101.3 | large | 35.37 | fine |
| 4 | 100 | 104.8 | middle | 14.39 | fine |
| 5 | 100 | 100.6 | small | 4.84 | coarse |
| 6 | 100 | 101.5 | middle | 14.85 | coarse |
| 7 | 100 | 101.5 | large | 35.30 | coarse |

*2.2. Transverse Distribution Model of Average Sediment Carrying Capacity*

After obtaining the transverse velocity distribution model of floodplain flow in curved compound channels, the sediment carrying capacity formula was introduced to calculate the vertical average sediment carrying capacity transverse distribution of the floodplain flow in curved compound channels.

We used the formula of Zhang Hongwu's sediment carrying capacity:

$$S_* = 2.5 \left[ \frac{(0.0022 + s_v)u^3}{k \frac{\gamma_s - \gamma_m}{\gamma_m} gh\omega_d} \ln \frac{H}{6d_{50}} \right]^{-0.62} \tag{1}$$

where $\kappa$ is the Karman constant, $\gamma_s$ is the bulk density of sediment, $h$ is total water depth, $g$ is acceleration of gravity, $H$ is depth of water, $u$ is vertical average downstream velocity, $s_v$ is bed slope, $d_{50}$ is median particle diameter, $\omega_d$ is average settling rate of bed sand, and $\gamma_m$ is the bulk density of muddy water. Therefore, once the lateral distribution of flow velocity and the lateral change of water depth are determined, the transverse distribution of sediment-carrying force can be solved directly.

*2.3. Theoretical Formula for Transverse Distribution of Water and Sediment*

In this study, SKM was used to predict the transverse distribution of the vertical average velocity in the compound channel, which did not consider the effect of vegetation at the beginning. Rameshwaran and Shiono [7] extended and improved the SKM and added vegetation factors to predict the velocity distribution of a straight channel with vegetation on beach. The results showed that this method can also be well applied to the prediction of velocity distribution of compound channels with vegetation on the beach.

Five calculation examples were used to analyze the transverse distribution of the velocity in the curved compound channel of the beach vegetation. The scenarios considered for the five calculation examples were (1) a curved compound channel with non-vegetation on the beach, (2) a curved compound channel with vegetation on the convex bank of the beach, (3) a curved compound channel with vegetation on the concave bank of the beach, (4) a curved compound channel with vegetation on both sides of the beach, and (5) a curved compound channel full of vegetation on the beach. Table 2 gives the governing equations and boundary conditions of the five cases, respectively.

**Table 2.** Control equations and boundary conditions of five examples.

| Example | Control Equations | Boundary Conditions | Parameters |
|---|---|---|---|
| (1) | $U_d^{(1)} = \left(A_1 e^{\gamma_1 y} + A_2 e^{-\gamma_1 y} + k_1\right)^{\frac{1}{2}}$ <br> $U_d^{(2)} = \left(\frac{8r}{r+y}\frac{\rho g H S_o - \Gamma_{mc}}{\rho f}\right)^{1/2}$ <br> $U_d^{(3)} = \left(A_3 e^{\gamma_3 y} + A_4 e^{-\gamma_3 y} + k_3\right)^{1/2}$ | $U_d^{(1)}\big|_{y=-B_1} = 0;\ U_d^{(1)}\big|_{y=0} = U_d^{(2)}\big|_{y=0};$ <br> $U_d^{(2)}\big|_{y=b} = U_d^{(3)}\big|_{y=b};\ U_d^{(3)}\big|_{y=b+B_2} = 0$ | $U_d$—vertical average downstream velocity; <br> $\rho$—fluid density; <br> $g$—acceleration of gravity; <br> $H$—depth of water; <br> $S_0$—bed slope in downstream direction; <br> $f$—bed surface friction factor; <br> $r$—available curvature of bed slope of main trough and beach; <br> $y$—horizontal coordinate; <br> $\Gamma$—secondary flow term; <br> $\gamma$—exponent of analytic solution of differential equation; <br> $A$—unknown constant; <br> $K$—empirical parameters. |
| (2) | $U_d^{(1)} = \left(A_1 e^{\gamma_1 y} + A_2 e^{-\gamma_1 y} + k_1\right)^{\frac{1}{2}}$ <br> $U_d^{(2)} = \left(\frac{8r}{r+y}\frac{\rho g H S_o - \Gamma_{mc}}{\rho f}\right)^{\frac{1}{2}}$ <br> $U_d^{(3)} = \left(A_3 e^{\gamma_3 y} + A_4 e^{-\gamma_3 y} + k_3\right)^{1/2}$ <br> $U_d^{(4)} = \left(A_5 e^{\gamma_4 y} + A_6 e^{-\gamma_4 y} + k_4\right)^{1/2}$ | $U_d^{(1)}\big|_{y=-B_1} = 0;\ U_d^{(1)}\big|_{y=0} = U_d^{(2)}\big|_{y=0};$ <br> $U_d^{(3)}\big|_{y=b+B_2} = U_d^{(4)}\big|_{y=b+B_2};$ <br> $\frac{\partial U_d^{(3)}}{\partial y}\big|_{y=b+B_2} = \frac{\partial U_d^{(4)}}{\partial y}\big|_{y=b+B_2};\ U_d^{(4)}\big|_{y=b+B_2+B_3} = 0$ | |
| (3) | $U_d^{(1)} = \left(A_1 e^{\gamma_1 y} + A_2 e^{-\gamma_1 y} + k_1\right)^{\frac{1}{2}}$ <br> $U_d^{(2)} = \left(A_3 e^{\gamma_2 y} + A_4 e^{-\gamma_2 y} + k_2\right)^{\frac{1}{2}}$ <br> $U_d^{(3)} = \left(\frac{8r}{r+y}\frac{\rho g H S_o - \Gamma_{mc}}{\rho f}\right)^{\frac{1}{2}}$ <br> $U_d^{(4)} = \left(A_5 e^{\gamma_4 y} + A_6 e^{-\gamma_4 y} + k_4\right)^{1/2}$ | $U_d^{(1)}\big|_{y=-B_1-B_2} = 0;\ U_d^{(1)}\big|_{y=-B_2} = U_d^{(2)}\big|_{y=-B_2}$ <br> $\frac{\partial U_d^{(2)}}{\partial y}\big|_{y=-B_2} = \frac{\partial U_d^{(3)}}{\partial y}\big|_{y=-B_2};\ U_d^{(2)}\big|_{y=0} = U_d^{(3)}\big|_{y=0};$ <br> $U_d^{(3)}\big|_{y=b} = U_d^{(4)}\big|_{y=b};$ <br> $U_d^{(4)}\big|_{y=b+B_3} = 0$ | |
| (4) | $U_d^{(1)} = \left(A_1 e^{\gamma_1 y} + A_2 e^{-\gamma_1 y} + k_1\right)^{\frac{1}{2}}$ <br> $U_d^{(2)} = \left(A_3 e^{\gamma_2 y} + A_4 e^{-\gamma_2 y} + k_2\right)^{\frac{1}{2}}$ <br> $U_d^{(3)} = \left(\frac{8r}{r+y}\frac{\rho g H S_o - \Gamma_{mc}}{\rho f}\right)^{\frac{1}{2}}$ <br> $U_d^{(4)} = \left(A_5 e^{\gamma_4 y} + A_6 e^{-\gamma_4 y} + k_4\right)^{\frac{1}{2}}$ <br> $U_d^{(5)} = \left(A_5 e^{\gamma_5 y} + A_6 e^{-\gamma_5 y} + k_5\right)^{1/2}$ | $U_d^{(1)}\big|_{y=-B_1-B_2} = 0;\ U_d^{(1)}\big|_{y=-B_2} = U_d^{(2)}\big|_{y=-B_2};$ <br> $\frac{\partial U_d^{(2)}}{\partial y}\big|_{y=-B_2} = \frac{\partial U_d^{(3)}}{\partial y}\big|_{y=-B_2};\ U_d^{(2)}\big|_{y=0} = U_d^{(3)}\big|_{y=0};$ <br> $U_d^{(3)}\big|_{y=b} = U_d^{(4)}\big|_{y=b};\ U_d^{(4)}\big|_{y=b+B_3} = U_d^{(5)}\big|_{y=b+B_3};$ <br> $\frac{\partial U_d^{(4)}}{\partial y}\big|_{y=b+B_3} = \frac{\partial U_d^{(5)}}{\partial y}\big|_{y=b+B_3};\ U_d^{(5)}\big|_{y=b+B_3+B_4} = 0$ | |
| (5) | $U_d^{(1)} = \left(A_1 e^{\gamma_1 y} + A_2 e^{-\gamma_1 y} + k_1\right)^{\frac{1}{2}}$ <br> $U_d^{(2)} = \left(\frac{8r}{r+y}\frac{\rho g H S_o - \Gamma_{mc}}{\rho f}\right)^{\frac{1}{2}}$ <br> $U_d^{(3)} = \left(A_3 e^{\gamma_3 y} + A_4 e^{-\gamma_3 y} + k_3\right)^{1/2}$ | $U_d^{(1)}\big|_{y=-B_1} = 0;\ U_d^{(1)}\big|_{y=0} = U_d^{(2)}\big|_{y=0};$ <br> $U_d^{(2)}\big|_{y=b} = U_d^{(3)}\big|_{y=b};\ U_d^{(3)}\big|_{y=b+B_2} = 0$ | |

## 3. Results

### 3.1. Transverse Distribution of Floodplain Flood Velocity in Meandering Compound Channels with Beach Vegetation

With the implementation of various test conditions, the velocity distribution continued to develop and change, which provided a reference for studying the erosion and deposition evolution of beach vegetation on curved compound channels. For the variation of velocity along the way, the transverse distribution of velocity in the upper section changed greatly, and the velocity in the lower section was relatively uniform. As a result, the deposition of beach land was mainly concentrated in the beach lip, the velocity of the middle and upper sections was not only concentrated in the main channel, but also appeared near the left and right bank walls, and even the velocity of the side wall was greater than that of the main channel. The beach vegetation had a very obvious effect on slowing down the flow velocity. The results showed that the arrangement of full vegetation on the beach had a uniform effect on the velocity distribution, which can reduce the phenomenon of excessive velocity on the side wall of the beach and increase the velocity in the main channel (Figure 3).

Due to the large number of test groups, this study only conducted a specific analysis for working condition 3. In a curved compound channel with no vegetation on the beach (CS6), the velocity of the beach on the left bank was less than that of the beach on the right, and the velocity of the main channel gradually increased from left to right. The maximum velocity appeared near the main trough on the right bank, and the maximum velocity was 0.22 m/s. The velocity near the right wall increased to 0.20 m/s. In a curved compound channel with vegetation on the convex bank of the beach (CS10), the velocity in the left bank was less than that in the right bank, and the velocity in the main channel was relatively small; the minimum velocity in the main channel was 0.050 m/s. The velocity of the vegetation area was the smallest in relation to nearby areas. The maximum velocity was 0.18 m/s on the right bank. In a curved compound channel with vegetation on the concave bank of the beach (CS14), the velocity of the left bank was less than that of the right bank, and the velocity of the main channel increased from left to right. The velocity of the vegetation area was significantly reduced compared with the nearby area, and the velocity was 0.026 m/s. The maximum flow velocity occurred on the right bank of the beach, and the maximum flow velocity was 0.12 m/s. In a curved compound channel with vegetation on both sides of the beach (CS18), the flow velocity was relatively uniform and fluctuated between 0.05 and 0.1 m/s, and the velocity of the vegetation area was reduced compared to nearby areas. In a curved compound channel with full vegetation on the beach (CS22), the flow velocity was relatively uniform, approximately 0.05 m/s, and the velocity of the main channel was slightly higher than that of the vegetation area (Figure 3B).

### 3.2. Theoretical Calculation of Transverse Distribution of Water and Sediment

3.2.1. Curved Compound Channel without Vegetation in the Beach

In the compound rectangular channel without vegetation, the velocity on the left bank was less than that on the right bank. The velocity of the main channel increased gradually from a concave to a convex bank. The calculation result of the model was relatively smoother than the measurement result on the right bank of the beach. The measured results showed that the velocity near the main channel on the right bank was relatively large, then gradually decreased towards the side wall, and the calculated results were not fully simulated. The sediment carrying capacity of the main channel changed greatly, and the concave bank was smaller than the convex bank. The sediment carrying capacity of the left bank was smaller than that of the right bank (Figure 4A).

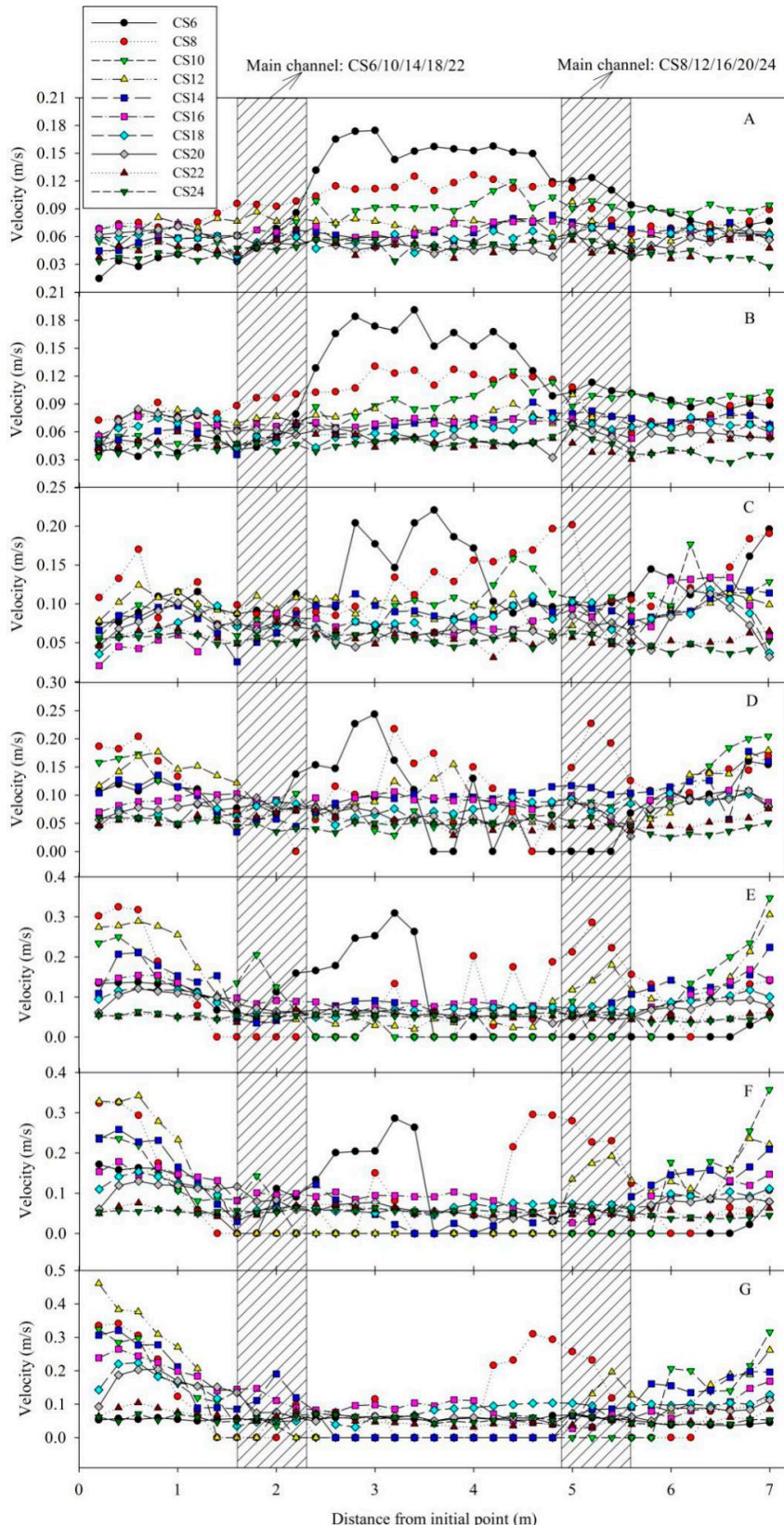

**Figure 3.** Transverse velocity distribution of each section under different working conditions, (**A**–**G**) represented conditions 1–7, respectively. The graphs correspond to the different conditions 1–7 described in Table 1.

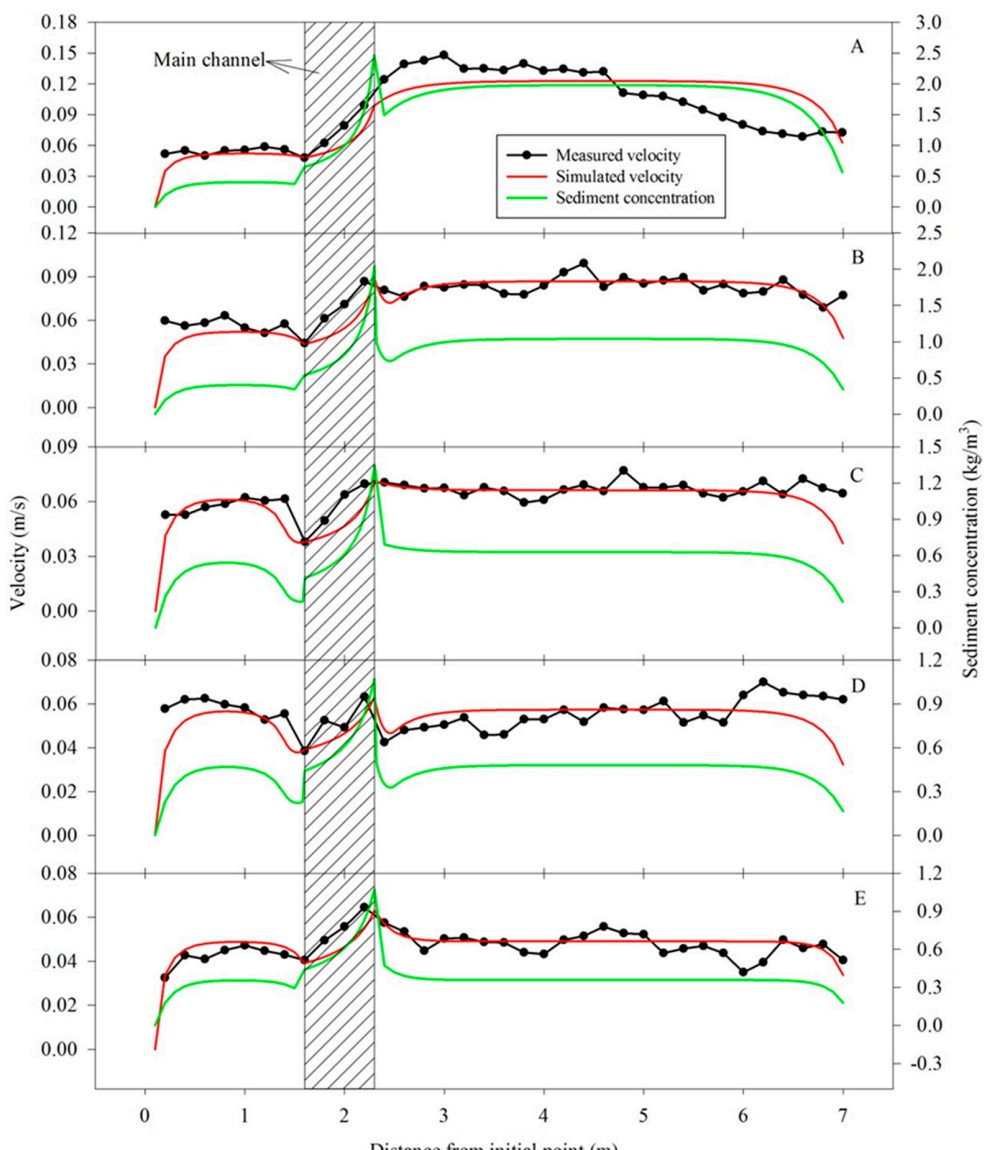

**Figure 4.** Velocity at the curved top of the curved compound rectangular channel and a comparison chart of the transverse distributions of sediment concentration. (**A**), curved compound channel with non-vegetation on the beach; (**B**), curved compound channel with vegetation on the convex bank of the beach; (**C**), curved compound channel with vegetation on the concave bank of the beach; (**D**), curved compound channel with vegetation on both sides of the beach; and (**E**), curved compound channel full of vegetation on the beach.

### 3.2.2. Curved Compound Channel with Vegetation on the Convex Bank of the Beach

In the compound rectangular channel with vegetation on the convex bank, the velocity on the left bank was less than that on the right bank. The velocity of the main channel increased gradually from a concave to a convex bank. There was a rapid decrease in the velocity in the vegetation area. The simulation results of the model and the measured data were relatively good. The sediment carrying capacity of the main channel changed greatly, and the concave bank was smaller than the convex bank. The sediment carrying capacity of the vegetation on the right bank had a sudden drop, and the sediment carrying capacity of the left bank was smaller than that of the right bank (Figure 4B).

### 3.2.3. Curved Compound Channel with Vegetation on the Concave Bank of the Beach

In the compound rectangular channel with vegetation on the concave bank, the velocity on the left bank was almost the same as that on the right bank. The velocity of the main channel increased gradually from a concave bank to a convex bank. There was a rapid decrease in the velocity in the vegetation area. The simulation results of the model and the measured data were relatively good. The sediment carrying capacity of the main channel changed greatly, and the concave bank was smaller than the convex bank. The sediment carrying capacity of the vegetation on the left bank decreased abruptly. The sediment carrying capacity of the left bank was similar to that of the right bank (Figure 4C).

### 3.2.4. Curved Compound Channel with Vegetation on Both Sides of the Beach

In the compound rectangular channel with vegetation on both sides of the beach, the flow velocity of the left and right beaches and the main channel was basically the same. The velocity of the main channel increased gradually from a concave bank to a convex bank. There was a rapid decrease in the velocity in the vegetation area on both banks. The simulation results of the model and the measured data were relatively good, only less than the measured data at the side wall. The sediment carrying capacity of the trough changed greatly, and the concave bank was smaller than the convex bank. There was a sudden drop in the sediment carrying capacity of the vegetation on both banks. The sediment carrying capacity of the left bank was similar to that of the right bank (Figure 4D).

### 3.2.5. Curved Compound Channel with Full Vegetation on the Beach

In the compound rectangular channel with full vegetation, the left and right banks were almost the same, and the velocity in the main channel was relatively large. The velocity of the main channel increased gradually from a concave bank to a convex bank. The calculated results of the model were in good agreement with the measured data. The sediment carrying capacity of the main channel was relatively large and varied greatly, and the concave bank was smaller than the convex bank. The sediment carrying capacity of the left bank was similar to that of the right bank (Figure 4E).

## 4. Discussions

The range of vegetation in the river is very wide, usually referring to soft and low blade grasses, dense shrubs, and hard trees. Vegetated river flow is a special and complex flow problem [22,23]. The emergence of vegetation has changed the internal structure of the original water flow to a large extent, increased the roughness of the beach, and affected the flood-carrying capacity of the entire river. Different vegetation conditions were set in this study, and it was found that beach vegetation slowed down the flow velocity obviously. Vegetation can stabilize the main channel. From the perspective of hydrodynamics, each tree is actually a very complex problem of circular flow around the cylinder [24]. The increase of resistance comes from the water resistance area of the tree and the resistance of the tree shape, while the "tree group" effect between the trees is more complex. In the current study, the beach is full of vegetation (tree group) and this has a uniform effect on the velocity distribution of the beach, which can reduce the excessive velocity of the beach side wall and increase the velocity effect of the main channel. In addition, the geomorphic environment of the channel is also unpredictable. For the compound section channel, there is a strong momentum exchange between the shoal and channel flows [25]. A strong turbulent vortex street is formed near the interface of the shoal and the channel. And the situation of vegetation on the shoal is more prominent and complex. The existence of vortex also hinders the smooth flow of water [26]. Therefore, for the biological revetment measures of riverbank vegetation, it is a new problem in the production practice to analyze its advantages and disadvantages, adapt measures to local conditions and plant rationally, which not only makes full use of riparian resources, maximizes its role in beach stabilization and embankment protection, but also guarantees flood discharge safety and protects the ecological environment [27].

For the flow characteristics of the curved compound channel without considering the effect of beach vegetation, many scholars have carried out experiments [28–31], numerical simulations [32–36], and theoretical research [37–39]. Among them, Lambert and Sellin [31] showed that the flow structure of a natural compound channel is extremely complex, and the traditional hydraulic analysis method cannot effectively estimate the large amount of momentum exchange at the interface between the main channel and floodplain. For the numerical simulation technology in the bend, Zarrati et al. [40] proposed a two-dimensional water depth average mathematical model to simulate the velocity and water depth at each transverse point of the section. There are relatively few studies on the flow characteristics under the coupling effect of beach vegetation and curved compound channel. Liu et al. [34] carried out the experiment in the compound curved channel with grass planted on the beach and obtained the experimental data of the velocity field, turbulence structure and Reynolds stress of the main channel. The experiment showed that the flexible grass on the beach had a significant reduction effect on the flood discharge capacity of the whole channel. Liu et al. [41] mainly studied the transverse velocity distribution in the curved compound channel with vegetation on the beach. They added the drag force to the momentum equation, considering the influence of the curvature of the main channel and the secondary flow. When integrating along the water depth, three different integration methods of the turbulent diffusion term were used to obtain three different forms of analytical solutions of velocity distribution in the water depth variation region. Compared with straight compound channel, the transverse distribution of flow in the curved compound channel was more complex. The influence of secondary flow in the curved channel was considered in the study of flow characteristics, and the research results were more in line with the characteristics of natural river, but the sediment problem of natural river was not considered. In the current study, the SKM method was used to analyze the force on the micro-control body of water flow. From the consideration of the momentum equation of lateral secondary flow inertia force and river curvature, and further considering the influence of beach vegetation, the transverse distribution model of the velocity of the compound channel of beach vegetation was established, and the transverse distribution law of the velocity of the compound channel of beach vegetation was proposed.

## 5. Conclusions

In this study, the model test of vegetation curved compound channel on beach land was carried out and abundant experimental data were obtained. The main stream is not only concentrated in the main channel, but also may appear near the foot of the levee on the left and right banks, resulting in flood flowing along the levee. Vegetation on the beach has an obvious slowing effect on the flow velocity. Whether the vegetation on both banks is in a row or planted with vegetation, it plays an important role in the stability of the main channel. Vegetation arrangement has a uniform effect on the velocity distribution of the beach, which can reduce the excessive velocity at the foot of the beach and increase the velocity effect of the main channel. Five numerical examples are used to test the lateral distribution model of the flow velocity of the curved compound channel. By comparing the results of the analytical solution model with the test results, the consistency between the velocity distribution obtained by the model and the test results was analyzed. On the whole, the analytical model is in good agreement with the experimental results.

**Author Contributions:** Conceptualization, M.Z. and S.J.; methodology, P.L.; validation, X.L.; formal analysis, M.Z.; investigation, Z.W.; resources, X.L.; data curation, A.W.; writing—original draft preparation, P.L.; writing—review and editing, S.J.; supervision, S.J.; project administration, P.L.; funding acquisition, M.Z. All authors contributed to the final version of the manuscript. All authors have read and agreed to the published version of the manuscript.

**Funding:** [National Key Research Priorities Program of China] grant number [2017YFC0404402], [National Natural Science Foundation of China] grant number [51809106], [Basic Research and Development Special Fund of Central Government for Non-profit Research Institutes] grant number [HKY-JBYW-2020-05].

**Institutional Review Board Statement:** No applicable.

**Informed Consent Statement:** No applicable.

**Data Availability Statement:** The data presented in this study is available on request from the corresponding author.

**Acknowledgments:** We would like to thank the potential reviewer very much for their valuable comments and suggestions. We also thank my other colleagues for their valuable comments and suggestions that have helped improve the manuscript.

**Conflicts of Interest:** The authors declare no conflict of interest.

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
