# Peer review of "Study on the Horizontal Distribution Law of Flood Water and Sediment Factors under the Effect of Vegetation on a Curved Beach"

_water, doi:10.3390/w13111441_

Round 1

Reviewer 1 Report

The work is interesting with the extensive experiment data and numerical results, though the manuscript seems not to be mature for the publication. Since the explanations for the figures are not sufficient especially in the results and discussion section, it is difficult to evaluate the novelty or advantage of the research works. Detailed comments are followings,

Line 93-99: The sentence is too long to understand the meaning.

Figure2: The figure is not clear to understand the layout of the vegetation along the channel.

Line 184-220: It is not clear the relationships of the explanations in the paragraph and the figures in Fig 3.

Figure3 and 5: What does the horizontal axis mean? Where is the initial point in the Fig2?

Line 225-244: How is this paragraph related to the present works in the discussion of the paper?

Line245-261: The paragraph explains the experimental results of measured suspended sediment distribution in detail, but it is difficult to get the message of the relationships with the vegetation effects.

Line349-355: The sentence is too long to understand the meaning.

Line 342-355: It is not clear the novelty or advantages of the research work.

Author Response

Reviewer #1

The work is interesting with the extensive experiment data and numerical results, though the manuscript seems not to be mature for the publication. Since the explanations for the figures are not sufficient especially in the results and discussion section, it is difficult to evaluate the novelty or advantage of the research works. Detailed comments are followings,

  • Line 93-99: The sentence is too long to understand the meaning.

A: We had revised the sentences as follow: In this study, a curved compound channel model of beach vegetation was established by means of experimental simulation, analysis of measured data and theoretical research. Based on the established model, the generalized model test of suspended mass floodplain flood evolution was carried out. The transverse distribution characteristics of flow and sediment factors in a curved compound channel under the influence of beach vegetation are analyzed, and an analytical solution model for the transverse distribution of velocity and sediment concentration was proposed.

  • Figure2: The figure is not clear to understand the layout of the vegetation along the channel.

A: We had revised Figure 2 and the figure caption as follow:

Figure 2. Experimental design. (CS5-CS8, curved compound channel with non-vegetation on the beach; CS9-CS12, curved compound channel with vegetation on the convex bank of the beach; CS13-CS16, curved compound channel with vegetation on the concave bank of the beach; CS17-CS20, curved compound channel with vegetation on both sides of the beach; CS21-CS24, curved compound channel full of vegetation on the beach)

  • Line 184-220: It is not clear the relationships of the explanations in the paragraph and the figures in Fig 3.

A: I'm sorry our description confused you. Because there are many test conditions, we only choose condition 3 for specific analysis. The test results of each working condition are summarized. We had revised the section as follow:

With the implementation of various test conditions, the velocity distribution continued to develop and change, which provided a reference for studying the erosion and deposition evolution of beach vegetation on curved compound channels. For the variation of velocity along the way, the transverse distribution of velocity in the upper section changed greatly, and the velocity in the lower section was relatively uniform. As a result that the deposition of beach land was mainly concentrated in the beach lip, the velocity of the middle and upper sections was not only concentrated in the main channel, but also appeared near the left and right bank walls, and even the velocity of the side wall was greater than that of the main channel. The beach vegetation had a very obvious effect on slowing down the flow velocity. The results showed that the arrangement of full vegetation on the beach had a uniform effect on the velocity distribution (CS22 and CS24), which can reduce the phenomenon of excessive velocity on the side wall of the beach and increase the velocity in the main channel (Figure 3).

Due to the large number of test groups, this study only conducted a specific analysis for working condition 3. In a curved compound channel with no vegetation on the beach (CS6), the velocity of the beach on the left bank was less than that of the beach on the right, and the velocity of the main channel gradually increased from left to right. The maximum velocity appeared near the main trough on the right bank, and the maximum velocity was 0.22 m/s. The velocity near the right wall increased to 0.20 m/s. In a curved compound channel with vegetation on the convex bank of the beach (CS10), the velocity in the left bank was less than that in the right bank, and the velocity in the main channel was relatively small; the minimum velocity in the main channel was 0.050 m/s. The velocity of the vegetation area was the smallest in relation to nearby areas. The maximum velocity was 0.18 m/s on the right bank. In a curved compound channel with vegetation on the concave bank of the beach (CS14), the velocity of the left bank was less than that of the right bank, and the velocity of the main channel increased from left to right. The velocity of the vegetation area was significantly reduced compared with the nearby area, and the velocity was 0.026 m/s. The maximum flow velocity occurred on the right bank of the beach, and the maximum flow velocity was 0.12 m/s. In a curved compound channel with vegetation on both sides of the beach (CS18), the flow velocity was relatively uniform and fluctuated between 0.05 and 0.1 m/s, and the velocity of the vegetation area was reduced compared to nearby areas. In a curved compound channel with full of vegetation on the beach (CS22), the flow velocity was relatively uniform, approximately 0.05 m/s, the velocity of the main channel was slightly higher than that of the vegetation area (Figure 3B).

  • Figure3 and 5: What does the horizontal axis mean? Where is the initial point in the Fig2?

A: Horizontal axis means the model width (That is, the total width of the transverse section, 7m, the following figure). The left bank was taken as the starting point, and the velocity monitoring points were set at intervals of 20cm.

    The left bank was taken as the starting point, and the velocity monitoring points were set at intervals of 20cm (the following figure).

  • Line 225-244: How is this paragraph related to the present works in the discussion of the paper?

A: We re-organized the manuscript, made the discussion as a separate part, and revised the discussion as follows:

The range of vegetation in the river is very wide, usually referring to soft and low blade grasses, dense shrubs and hard trees. Vegetated river flow is a special and complex flow problem [22, 23]. The emergence of vegetation has changed the internal structure of the original water flow to a large extent, increased the roughness of the beach, and affected the flood carrying capacity of the entire river. From the perspective of hydrodynamics, each tree is actually a very complex problem of circular flow around the cylinder [24]. The increase of resistance comes from the water resistance area of the tree and the resistance of the tree shape, while the "tree group" effect between the trees is more complex. In addition, the geomorphic environment of the channel is also unpredictable. For the compound section channel, there is a strong momentum exchange between the shoal and channel flows [25]. A strong turbulent vortex street is formed near the interface of the shoal and channel. And the situation of vegetation on the shoal is more prominent and complex. The existence of vortex also hinders the smooth flow of water [26]. Therefore, for the biological revetment measures of riverbank vegetation, it is a new problem in the production practice to analyze its advantages and disadvantages, adapt measures to local conditions and plant rationally, which not only makes full use of riparian resources, maximizes its role in beach stabilization and embankment protection, but also guarantees flood discharge safety and protects the ecological environment [27].

For the flow characteristics of the curved compound channel without considering the effect of beach vegetation, many scholars have carried out experiments [28-31], numerical simulations [32-36] and theoretical research [37-39]. Among them, Lambert and Sellin [31] showed that the flow structure of a natural compound channel is extremely complex, and the traditional hydraulic analysis method cannot effectively estimate the large amount of momentum exchange at the interface between the main channel and floodplain. For the numerical simulation technology in the bend, Zarrati et al. [40] proposed a two-dimensional water depth average mathematical model to simulate the velocity and water depth at each transverse point of the section. There are relatively few studies on the flow characteristics under the coupling effect of beach vegetation and curved compound channel. Liu et al. [34] carried out the experiment in the compound curved channel with grass planted on the beach and obtained the experimental data of the velocity field, turbulence structure and Reynolds stress of the main channel. The experiment showed that the flexible grass on the beach had a significant reduction effect on the flood discharge capacity of the whole channel. Liu et al. [41] mainly studied the transverse velocity distribution in the curved compound channel with vegetation on the beach. They added the drag force to the momentum equation, considering the influence of the curvature of the main channel and the secondary flow. When integrating along the water depth, three different integration methods of the turbulent diffusion term were used to obtain three different forms of analytical solutions of velocity distribution in the water depth variation region. Compared with straight compound channel, the transverse distribution of flow in the curved compound channel was more complex. The influence of secondary flow in the curved channel was considered in the study of flow characteristics, and the research results were more in line with the characteristics of natural river, but the sediment problem of natural river was not considered.

  • Line245-261: The paragraph explains the experimental results of measured suspended sediment distribution in detail, but it is difficult to get the message of the relationships with the vegetation effects.

A: We had deleted the section 2.2.

  • Line349-355: The sentence is too long to understand the meaning.

A: We had revised the conclusions as follow:

    In this study, the model test of vegetation curved compound channel on beach land was carried out and abundant experimental data were obtained. The main stream is not only concentrated in the main channel, but also may appear near the foot of the levee on the left and right banks, resulting in flood flowing along the levee. Vegetation on the beach has an obvious slowing effect on the flow velocity. Whether the vegetation on both banks is in a row or planted with vegetation, it plays an important role in the stability of the main channel. Vegetation arrangement has a uniform effect on the velocity distribution of the beach, which can reduce the excessive velocity at the foot of the beach and increase the velocity effect of the main channel. Five numerical examples are used to test the lateral distribution model of the flow velocity of the curved compound channel. By comparing the results of the analytical solution model with the test results, the consistency between the velocity distribution obtained by the model and the test results was analyzed. On the whole, the analytical model is in good agreement with the experimental results.

  • Line 342-355: It is not clear the novelty or advantages of the research work.

A: We had revised the conclusions as follow:

    In this study, the model test of vegetation curved compound channel on beach land was carried out and abundant experimental data were obtained. The main stream is not only concentrated in the main channel, but also may appear near the foot of the levee on the left and right banks, resulting in flood flowing along the levee. Vegetation on the beach has an obvious slowing effect on the flow velocity. Whether the vegetation on both banks is in a row or planted with vegetation, it plays an important role in the stability of the main channel. Vegetation arrangement has a uniform effect on the velocity distribution of the beach, which can reduce the excessive velocity at the foot of the beach and increase the velocity effect of the main channel. Five numerical examples are used to test the lateral distribution model of the flow velocity of the curved compound channel. By comparing the results of the analytical solution model with the test results, the consistency between the velocity distribution obtained by the model and the test results was analyzed. On the whole, the analytical model is in good agreement with the experimental results.

Reviewer 2 Report

Mingwu and colleagues provide a generalized model based on a combination of experimental simulations and theoretical research for assessing the floodplain food evolution, including the horizontal distribution of flood water and the related sediment factors, under the effect of vegetation on a curved beach. Generally, the topic of the manuscript is of international interest, the work is important, the analysis and methodology are clear enough, and the manuscript is generally well-structured. More explicitly, all data are sufficient, and the adopted methods are appropriated, as well as the treatment of the data. However, some figures need improvement (please see my comments below). The length of the paper is appropriated for this journal, with all interpretations and conclusions to be in general well justified. The text is also well organized; however, results and discussion sections should be distinguished, and this will make the manuscript easily readable and understandable. Finally the bibliography is well updated without self-citations. The English is in relatively good shape, but some places need some improvements (please see my comments below). Overall, I have a couple of significant comments and suggestions, and therefore ask for revision (minor) before accepting this manuscript for Water. So, please take them into account in order this promising contribution to be publishable. The manuscript is acceptable with minor revision.

Minor Comments and Suggestions:

-L58-79: The whole paragraph seems like a chapter book related to the methods applied for the vegetation factors. However, this is not the case here. If the authors want to keep them into the manuscript, they should create a new section entitle “Background work” or “Applied methodology” after the Introduction.

-L80-82: Rephrase. It is not clear enough.

-L84-85: Rephrase. The word “characteristics” is redundant by referring it 3 times within the same sentence

-L90-92: Rephrase again

-L93-99: By writing so large sentences, the meaning is missing. This is a very good example. Split it into 2 or 3 different sentences.

-L99-102: Possibly, it should be better if the authors add the general usage of the proposed model and not only highlight its regional character and application.

-L138-149: What exactly do the authors mean by mentioning “to consider different sediment concentrations”? This should be clarified into the text by a better way

-L156-157: The specific reason of the selection of that equation instead of others should be added here

General comment: Results should be written apart from the discussion. The authors should be re-written this part of the manuscript by distinguishing the results from the discussion sections.

-L207: Rephrase it is not clear

-Figure 3 is very busy and difficult to understand by the potential reader. Maybe the usage of different colors for the different records would be helpful.

-L227-228: Which are exactly the development of new technologies and the advanced measuring instruments? Briefly refer to them both

-L245-261: Section 3.2. should be removed to the results during the general re-organization of the manuscript

-Conclusions: The findings of this work should be better described at this section. I propose this section to be re-written from the beginning highlighting to the most significant observations

Author Response

Reviewer #2

Mingwu and colleagues provide a generalized model based on a combination of experimental simulations and theoretical research for assessing the floodplain food evolution, including the horizontal distribution of flood water and the related sediment factors, under the effect of vegetation on a curved beach. Generally, the topic of the manuscript is of international interest, the work is important, the analysis and methodology are clear enough, and the manuscript is generally well-structured. More explicitly, all data are sufficient, and the adopted methods are appropriated, as well as the treatment of the data. However, some figures need improvement (please see my comments below). The length of the paper is appropriated for this journal, with all interpretations and conclusions to be in general well justified. The text is also well organized; however, results and discussion sections should be distinguished, and this will make the manuscript easily readable and understandable. Finally the bibliography is well updated without self-citations. The English is in relatively good shape, but some places need some improvements (please see my comments below). Overall, I have a couple of significant comments and suggestions, and therefore ask for revision (minor) before accepting this manuscript for Water. So, please take them into account in order this promising contribution to be publishable. The manuscript is acceptable with minor revision.

Minor Comments and Suggestions:

  • L58-79: The whole paragraph seems like a chapter book related to the methods applied for the vegetation factors. However, this is not the case here. If the authors want to keep them into the manuscript, they should create a new section entitle “Background work” or “Applied methodology” after the Introduction.

 A: We had deleted the paragraph, and added the sentence as follow in the introduction:

 Although the existing research has some limitations, it also proves that under certain environment, the influence of the growth of riparian vegetation on the flow resistance is smaller than previously thought [11,12]. Due to the limitation of field measurement, the research on the flow structure of river with vegetation is mainly carried out in indoor flume. The existing indoor research results show that the flow structure in the vegetated area is far more complex than we realize [13]. Due to the limitation of the previous measuring instruments, the early research mainly focused on the estimation of the average velocity, resistance law and roughness coefficient, and the flow structure in the vegetation area has not been described, analyzed and understood in detail [14]. The significance of these hydrodynamic characteristics in flood control and sediment movement is not fully studied. For the compound section channel with vegetation on the beach, although the surface shear stress generated by the momentum exchange between the beach and channel flows has been calculated by most researchers on the basis of the unbalanced force, the technology of measuring the surface shear stress directly by using pulsating function is not perfect [15-17]. In addition, the method of quantifying the momentum transfer at the interface between the main channel and the beach by using measurable parameters and considering the surface shear stress of the beach vegetation needs to be developed [18]. Up to now, the experimental study of flow in vegetated channel, whether it is single section channel or compound section channel, is only limited to straight channel, without considering the influence of section shape on flow [19-21]. Therefore, it is necessary to study the bend flow with vegetation.

  • L80-82: Rephrase. It is not clear enough.

A: We had revised the sentences as follow: The research on the structure of floodplain water and sediment under the effect of vegetation in the meandering compound channel was mainly focused on the movement law of clear water flow and bed load.

  • L84-85: Rephrase. The word “characteristics” is redundant by referring it 3 times within the same sentence

A: We had revised the sentences as follow: As for sandy rivers, sediment-laden flow is one of the characteristics that cannot be described by the existing clear water flow and bed load movement rules in terms of physical, movement and sediment transport characteristics.

  • L90-92: Rephrase again

A: We deleted the sentences.

  • L93-99: By writing so large sentences, the meaning is missing. This is a very good example. Split it into 2 or 3 different sentences.

A: We had revised the sentences as follow: In this study, a curved compound channel model of beach vegetation was established by means of experimental simulation, analysis of measured data and theoretical research. Based on the established model, the generalized model test of suspended mass floodplain flood evolution was carried out. The transverse distribution characteristics of flow and sediment factors in a curved compound channel under the influence of beach vegetation are analyzed, and an analytical solution model for the transverse distribution of velocity and sediment concentration was proposed.

  • L99-102: Possibly, it should be better if the authors add the general usage of the proposed model and not only highlight its regional character and application.

A: We had revised the sentences as follow: This achievement will provide the theoretical basis for beach area application and river regulation and has great significance for enriching the basic theory of water and sediment movement, promoting the integration of hydraulics, river dynamics and ecology.

  • L138-149: What exactly do the authors mean by mentioning “to consider different sediment concentrations”? This should be clarified into the text by a better way

A: We had revised the sentences as follow: In order to study the flow and sediment situation of curved compound channels with vegetation on the beach more comprehensively, seven types of working conditions were designed through different combinations of sediment concentration and sediment particle size. The design flow was 100 m3/h, and the actual situation was a little different. Condition 0 was the initial condition to adapt to the flow conditions of the initial design terrain; condition 1 was under the condition of clear water; conditions 2–4 were under the condition of relatively fine sediment; conditions 5–7 were under the condition of relatively coarse sediment. The different sediment concentrations were approximately 5 kg/m3 for small sediment concentrations, 14.5 kg/m3 for medium sediment concentrations and 35.3 kg/m3 for large sediment concentrations (Table 1). The test tailgate can auto matically adjust the height and control of the water level of the tailgate.

(8) L156-157: The specific reason of the selection of that equation instead of others should be added here

A: We had revised the sentences as follow: We used the formula of Zhang Hongwu’s sediment carrying capacity.

(9) General comment: Results should be written apart from the discussion. The authors should be re-written this part of the manuscript by distinguishing the results from the discussion sections.

A: we had distinguished the results from the discussion sections in the manuscript. The revised discussions are as follow:

The range of vegetation in the river is very wide, usually referring to soft and low blade grasses, dense shrubs and hard trees. Vegetated river flow is a special and complex flow problem [22, 23]. The emergence of vegetation has changed the internal structure of the original water flow to a large extent, increased the roughness of the beach, and affected the flood carrying capacity of the entire river. From the perspective of hydrodynamics, each tree is actually a very complex problem of circular flow around the cylinder [24]. The increase of resistance comes from the water resistance area of the tree and the resistance of the tree shape, while the "tree group" effect between the trees is more complex. In addition, the geomorphic environment of the channel is also unpredictable. For the compound section channel, there is a strong momentum exchange between the shoal and channel flows [25]. A strong turbulent vortex street is formed near the interface of the shoal and channel. And the situation of vegetation on the shoal is more prominent and complex. The existence of vortex also hinders the smooth flow of water [26]. Therefore, for the biological revetment measures of riverbank vegetation, it is a new problem in the production practice to analyze its advantages and disadvantages, adapt measures to local conditions and plant rationally, which not only makes full use of riparian resources, maximizes its role in beach stabilization and embankment protection, but also guarantees flood discharge safety and protects the ecological environment [27].

For the flow characteristics of the curved compound channel without considering the effect of beach vegetation, many scholars have carried out experiments [28-31], numerical simulations [32-36] and theoretical research [37-39]. Among them, Lambert and Sellin [31] showed that the flow structure of a natural compound channel is extremely complex, and the traditional hydraulic analysis method cannot effectively estimate the large amount of momentum exchange at the interface between the main channel and floodplain. For the numerical simulation technology in the bend, Zarrati et al. [40] proposed a two-dimensional water depth average mathematical model to simulate the velocity and water depth at each transverse point of the section. There are relatively few studies on the flow characteristics under the coupling effect of beach vegetation and curved compound channel. Liu et al. [34] carried out the experiment in the compound curved channel with grass planted on the beach and obtained the experimental data of the velocity field, turbulence structure and Reynolds stress of the main channel. The experiment showed that the flexible grass on the beach had a significant reduction effect on the flood discharge capacity of the whole channel. Liu et al. [41] mainly studied the transverse velocity distribution in the curved compound channel with vegetation on the beach. They added the drag force to the momentum equation, considering the influence of the curvature of the main channel and the secondary flow. When integrating along the water depth, three different integration methods of the turbulent diffusion term were used to obtain three different forms of analytical solutions of velocity distribution in the water depth variation region. Compared with straight compound channel, the transverse distribution of flow in the curved compound channel was more complex. The influence of secondary flow in the curved channel was considered in the study of flow characteristics, and the research results were more in line with the characteristics of natural river, but the sediment problem of natural river was not considered.

(10) L207: Rephrase it is not clear

 A: We had revised the sentences as follow: With the implementation of various test conditions.

(11) Figure 3 is very busy and difficult to understand by the potential reader. Maybe the usage of different colors for the different records would be helpful.

A: We had revised the Figure 3 as follow:

(12) L227-228: Which are exactly the development of new technologies and the advanced measuring instruments? Briefly refer to them both

 A: we had deleted the paragraph, and re-written the results and discussions.

(13) L245-261: Section 3.2. should be removed to the results during the general re-organization of the manuscript

 A: we had deleted the secion 3.2 in the manuscript.

(14) Conclusions: The findings of this work should be better described at this section. I propose this section to be re-written from the beginning highlighting to the most significant observations

A: We had revised the conclusions as follow:

In this study, the model test of vegetation curved compound channel on beach land was carried out and abundant experimental data were obtained. The main stream is not only concentrated in the main channel, but also may appear near the foot of the levee on the left and right banks, resulting in flood flowing along the levee. Vegetation on the beach has an obvious slowing effect on the flow velocity. Whether the vegetation on both banks is in a row or planted with vegetation, it plays an important role in the stability of the main channel. Vegetation arrangement has a uniform effect on the velocity distribution of the beach, which can reduce the excessive velocity at the foot of the beach and increase the velocity effect of the main channel. Five numerical examples are used to test the lateral distribution model of the flow velocity of the curved compound channel. By comparing the results of the analytical solution model with the test results, the consistency between the velocity distribution obtained by the model and the test results was analyzed. On the whole, the analytical model is in good agreement with the experimental results.

Reviewer 3 Report

Dear author,

The paper is focused on the combination of experimental simulation and theoretical research to carry out a generalized model test of floodplain flood evolution. The authors analysed the distribution characteristics of sediment-laden flow and sediment factors in a curved compound channel under the conditions of beach vegetation. The results provide the theoretical basis for beach area application and river regulation in the lower Yellow River.

In my opinion the strengths of this research are:

  1. combination of experimental simulation (actual measurement data analysis in water tank model) and theoretical research using Shiono and Knight method to predict the distribution of the vertical average velocity in the compound channel
  2. modelling five scenarios of vegetation
  3. application of extended SKM model in curved channel (with 5 scenarios of vegetation)
  4. the significance of results for enriching the theory of water and sediment movement on a curved beach
  5. the object of the article is appropriate for this journal

The weaknesses are (recommendations for improving the quality of paper):

  1. the quality of some figures (e.g. fig. 1)
  2. the information in introduction is not sufficiently supported by references (only 14 references)
  3. the main results are missing in the conclusion

My recommendations for improving the quality of paper are given in paragraph 3 (weaknesses).

Author Response

Reviewer #3

The paper is focused on the combination of experimental simulation and theoretical research to carry out a generalized model test of floodplain flood evolution. The authors analysed the distribution characteristics of sediment-laden flow and sediment factors in a curved compound channel under the conditions of beach vegetation. The results provide the theoretical basis for beach area application and river regulation in the lower Yellow River.In my opinion the strengths of this research are:

combination of experimental simulation (actual measurement data analysis in water tank model) and theoretical research using Shiono and Knight method to predict the distribution of the vertical average velocity in the compound channel modelling five scenarios of vegetation application of extended SKM model in curved channel (with 5 scenarios of vegetation) the significance of results for enriching the theory of water and sediment movement on a curved beach the object of the article is appropriate for this journal The weaknesses are (recommendations for improving the quality of paper):

(1) the quality of some figures (e.g. fig. 1)

A: We had revised the Figure 1 and Figure 3 and improved the quality in the manuscript.

  • the information in introduction is not sufficiently supported by references (only 14 references)

A: We had revised the introduction, and added the references in the introduction.

(3) the main results are missing in the conclusion

A: We had revised the conclusion as follow:

In this study, the model test of vegetation curved compound channel on beach land was carried out and abundant experimental data were obtained. The main stream is not only concentrated in the main channel, but also may appear near the foot of the levee on the left and right banks, resulting in flood flowing along the levee. Vegetation on the beach has an obvious slowing effect on the flow velocity. Whether the vegetation on both banks is in a row or planted with vegetation, it plays an important role in the stability of the main channel. Vegetation arrangement has a uniform effect on the velocity distribution of the beach, which can reduce the excessive velocity at the foot of the beach and increase the velocity effect of the main channel. Five numerical examples are used to test the lateral distribution model of the flow velocity of the curved compound channel. By comparing the results of the analytical solution model with the test results, the consistency between the velocity distribution obtained by the model and the test results was analyzed. On the whole, the analytical model is in good agreement with the experimental results.

  • My recommendations for improving the quality of paper are given in paragraph 3 (weaknesses).

A: We had deleted paragraph 3, and added the sentences in the introduction as follow:

Although the existing research has some limitations, it also proves that under certain environment, the influence of the growth of riparian vegetation on the flow resistance is smaller than previously thought [11,12]. Due to the limitation of field measurement, the research on the flow structure of river with vegetation is mainly carried out in indoor flume. The existing indoor research results show that the flow structure in the vegetated area is far more complex than we realize [13]. Due to the limitation of the previous measuring instruments, the early research mainly focused on the estimation of the average velocity, resistance law and roughness coefficient, and the flow structure in the vegetation area has not been described, analyzed and understood in detail [14]. The significance of these hydrodynamic characteristics in flood control and sediment movement is not fully studied. For the compound section channel with vegetation on the beach, although the surface shear stress generated by the momentum exchange between the beach and channel flows has been calculated by most researchers on the basis of the unbalanced force, the technology of measuring the surface shear stress directly by using pulsating function is not perfect [15-17]. In addition, the method of quantifying the momentum transfer at the interface between the main channel and the beach by using measurable parameters and considering the surface shear stress of the beach vegetation needs to be developed [18]. Up to now, the experimental study of flow in vegetated channel, whether it is single section channel or compound section channel, is only limited to straight channel, without considering the influence of section shape on flow [19-21]. Therefore, it is necessary to study the bend flow with vegetation.

Round 2

Reviewer 1 Report

The manuscript has been well revised with clear figures and text sentences,  responding appropriately to the all reviewers comments. The structure of the manuscript is also well re-organized by the separation of the results and discussion session in the revised version. 

Only one general comment for Line 281-323. It should be better if the comments were included  in the discussion section about the findings obtained through the the present research in comparison with the mentioned referenced works in the section. 

Author Response

The manuscript has been well revised with clear figures and text sentences,  responding appropriately to the all reviewers comments. The structure of the manuscript is also well re-organized by the separation of the results and discussion session in the revised version. 

Only one general comment for Line 281-323. It should be better if the comments were included  in the discussion section about the findings obtained through the the present research in comparison with the mentioned referenced works in the section. 

A: We had revised the discussion as follow:

The range of vegetation in the river is very wide, usually referring to soft and low blade grasses, dense shrubs and hard trees. Vegetated river flow is a special and complex flow problem [22, 23]. The emergence of vegetation has changed the internal structure of the original water flow to a large extent, increased the roughness of the beach, and affected the flood carrying capacity of the entire river. Different vegetation conditions were set in this study, and it was found that beach vegetation slowed down the flow velocity obviously. Vegetation can stabilize the main channel. From the perspective of hydrodynamics, each tree is actually a very complex problem of circular flow around the cylinder [24]. The increase of resistance comes from the water resistance area of the tree and the resistance of the tree shape, while the "tree group" effect between the trees is more complex. In the current study, the beach is full of vegetation (tree group) has a uniform effect on the velocity distribution of the beach, which can reduce the excessive velocity of the beach side wall and increase the velocity effect of the main channel. In addition, the geomorphic environment of the channel is also unpredictable. For the compound section channel, there is a strong momentum exchange between the shoal and channel flows [25]. A strong turbulent vortex street is formed near the interface of the shoal and channel. And the situation of vegetation on the shoal is more prominent and complex. The existence of vortex also hinders the smooth flow of water [26]. Therefore, for the biological revetment measures of riverbank vegetation, it is a new problem in the production practice to analyze its advantages and disadvantages, adapt measures to local conditions and plant rationally, which not only makes full use of riparian resources, maximizes its role in beach stabilization and embankment protection, but also guarantees flood discharge safety and protects the ecological environment [27].

For the flow characteristics of the curved compound channel without considering the effect of beach vegetation, many scholars have carried out experiments [28–31], numerical simulations [32–36] and theoretical research [37–39]. Among them, Lambert and Sellin [31] showed that the flow structure of a natural compound channel is extremely complex, and the traditional hydraulic analysis method cannot effectively estimate the large amount of momentum exchange at the interface between the main channel and floodplain. For the numerical simulation technology in the bend, Zarrati et al. [40] proposed a two-dimensional water depth average mathematical model to simulate the velocity and water depth at each transverse point of the section. There are relatively few studies on the flow characteristics under the coupling effect of beach vegetation and curved compound channel. Liu et al. [34] carried out the experiment in the compound curved channel with grass planted on the beach and obtained the experimental data of the velocity field, turbulence structure and Reynolds stress of the main channel. The experiment showed that the flexible grass on the beach had a significant reduction effect on the flood discharge capacity of the whole channel. Liu et al. [41] mainly studied the transverse velocity distribution in the curved compound channel with vegetation on the beach. They added the drag force to the momentum equation, considering the influence of the curvature of the main channel and the secondary flow. When integrating along the water depth, three different integration methods of the turbulent diffusion term were used to obtain three different forms of analytical solutions of velocity distribution in the water depth variation region. Compared with straight compound channel, the transverse distribution of flow in the curved compound channel was more complex. The influence of secondary flow in the curved channel was considered in the study of flow characteristics, and the research results were more in line with the characteristics of natural river, but the sediment problem of natural river was not considered. In the current study, the SKM method was used to analyze the force on the micro-control body of water flow. From the consideration of the momentum equation of lateral secondary flow inertia force and river curvature, and further considering the influence of beach vegetation, the transverse distribution model of the velocity of the compound channel of beach vegetation was established, and the transverse distribution law of the velocity of the compound channel of beach vegetation was proposed.
